# Behavioural Determinants of COVID-19-Vaccine Acceptance in Rural Areas of Six Lower- and Middle-Income Countries

**DOI:** 10.3390/vaccines10020214

**Published:** 2022-01-29

**Authors:** Thomas P. Davis, Adugna Kebede Yimam, Md Abul Kalam, Asrat Dibaba Tolossa, Robert Kanwagi, Sarah Bauler, Loria Kulathungam, Heidi Larson

**Affiliations:** 1World Vision International, 1202 Geneva, Switzerland; Adugna_Kebede@wvi.org (A.K.Y.); Rkanwagi@yahoo.com (R.K.); Sarah_Bauler@wvi.org (S.B.); Loria_Kulathungam@wvi.org (L.K.); 2Bangladesh Country Office, Helen Keller International, Dhaka 1212, Bangladesh; a.kalam724@gmail.com; 3World Vision Canada, Mississauga, ON L5T 2Y4, Canada; Asrat_Dibaba@worldvision.ca; 4Vaccine Confidence Project, London School of Hygiene and Tropical Medicine, London WC1E 7HT, UK; Heidi.Larson@LSHTM.ac.uk

**Keywords:** COVID-19, vaccine, hesitancy, acceptance, behaviour, determinants

## Abstract

Delayed acceptance or refusal of COVID-19 vaccines may increase and prolong the threat to global public health and the economy. Identifying behavioural determinants is considered a critical step in explaining and addressing the barriers of vaccine refusal. This study aimed to identify the behavioural determinants of COVID-19-vaccine acceptance and provide recommendations to design actionable interventions to increase uptake of the COVID-19 vaccine in six lower- and middle-income countries. Taking into consideration the health belief model and the theory of reasoned action, a barrier analysis approach was employed to examine twelve potential behavioural determinants of vaccine acceptance in Bangladesh, India, Myanmar, Kenya, the Democratic Republic of the Congo (DRC), and Tanzania. In all six countries, at least 45 interviews with those who intended to get the vaccine (“Acceptors”) and another 45 or more interviews with those who did not (“Non-acceptors”) were conducted, totalling 542 interviews. Data analysis was performed to find statistically significant (*p* < 0.05) differences between Acceptors and Non-acceptors of COVID-19 vaccines and to identify which beliefs were most highly associated with acceptance and non-acceptance of vaccination based on the estimated relative risk. The analysis showed that perceived social norms, perceived positive and negative consequences, perceived risk, perceived severity, trust, perceived safety, and expected access to COVID-19 vaccines had the highest associations with COVID-19-vaccine acceptance in Bangladesh, Kenya, Tanzania, and the DRC. Additional behavioural determinants found to be significant in Myanmar and India were perceived self-efficacy, trust in COVID-19 information provided by leaders, perceived divine will, and perceived action efficacy of the COVID-19 vaccines. Many of the determinants were found to be significant, and their level of significance varied from country to country. National and local plans should include messages and activities that address the behavioural determinants found in this study to significantly increase the uptake of COVID-19 vaccines across these countries.

## 1. Introduction

As of 23 December 2021, over 275 million people have become infected by the COVID-19 virus, and over 5.3 million people have died globally [1]. Due to the highly infectious Delta and Omicron variants, the pandemic is expected to continue to impose significant burdens of morbidity and mortality and to severely disrupt societies and economies. The introduction of effective COVID-19 vaccines is the only clinical preventive measure. While these vaccines have been a positive development in curbing the pandemic’s effect, anecdotal reports and some studies have shown that there is hesitancy to take the vaccine among many populations. The ability of the COVID-19 vaccines to stop the spread of the virus will depend on both the efficacy of the vaccines and the degree to which people are willing to get one of these vaccines—vaccine acceptance. 

Vaccine hesitancy or refusal of COVID-19 vaccines is a growing concern worldwide. In one multi-country survey, only 71.5% of participants reported that they would be very or somewhat likely to take a COVID-19 vaccine [2]. A UK national representative survey (12–18 December 2020) revealed that trust was a key predictor of vaccine hesitancy. Low perceived personal threat, fears around accelerated vaccine development, concerns about side effects, misunderstandings regarding herd immunity, and beliefs that the virus is manmade or will be used for population control all may contribute to the likelihood of vaccine hesitancy [3]. A rapid systematic review around COVID-19 vaccine hesitancy in 31 countries found that perceived risk, concerns over vaccine safety and effectiveness, doctors’ recommendations, political-party orientation, and inoculation history were common factors [4]. Importantly, unveiling the underlying key social and behavioural determinants is a cornerstone in promoting COVID-19 vaccination. A growing number of studies involved with investigating the willingness to be vaccinated with COVID-19 vaccines showed the propagation of anti-vaccination movements in several countries [5,6]. Other studies have identified demographic, socioeconomic, and behavioural factors that are linked with vaccine acceptance [7,8,9,10]. The influential factors include gender, age, and marital status [11]; educational attainment and ethnic origin [12,13]; psychological factors and beliefs [14]; and previous vaccination history with influenza and other vaccines [15]. Besides, concerns over the efficacy and safety of vaccines, misinformation, and reduced levels of public trust in vaccines are thought to be responsible for vaccine-related risk among the general public [2,16,17,18].

Behavioural studies on other vaccines have shown that the decision to vaccinate is often based on factors that are the focus on the Health Belief Model (HBM)—such as perceived benefits, perceived effectiveness of the vaccine (also known as perceived action efficacy), perceived vaccine side-effects, and reduced perceived risk of infection by COVID-19 [19]. Systematic reviews on behavioural determinants have shown that assessing determinants that are included in the HBM revealed significant determinants associated with the acceptance of human papillomavirus [20] and influenza-vaccination uptake [21]. Similarly, the theory of planned behaviour, which was developed from the theory of reasoned action, showed promising findings in capturing subjective norms, attitudes, and perceived behavioural control regarding vaccinations [22]. Moreover, trust, misconceptions, misinformation, and lack of knowledge among the community on vaccine-preventable diseases are considered influential determinants of the lower level of acceptance [2,4,9,23,24]. These factors have influenced vaccine uptake during previous pandemics and outbreaks caused by H1N1, MERS, SARS, and Ebola virus [2,4,9,24]. A barrier analysis study in Dhaka, Bangladesh found that perceived social norms, perceived safety of COVID-19 vaccines and trust in them, perceived risk/susceptibility, perceived self-efficacy, perceived positive and negative consequences, perceived action efficacy, perceived severity of COVID-19, access, and perceived divine will were significant predictors of vaccine acceptance among an urban population in Bangladesh [8].

Vaccine hesitancy and acceptance are complex in nature, and vaccine decisions can vary according to context, time, and place [25], so many earlier global studies that focused on demographic determinants may have limited value in understanding the determinants of COVID-19-vaccine acceptance in a given country, time, and culture. Understanding how different behavioural attributes affect individual preferences about vaccination at a granular level can help inform public health authorities about the actionable activities and messages necessary to achieve broader community uptake of COVID-19 vaccines. Therefore, the primary objective of this study was to identify the behavioural determinants of COVID-19 vaccines in Bangladesh, the Democratic Republic of the Congo (DRC), India, Kenya, Myanmar, and Tanzania for use by different stakeholders. This study will also help practitioners to design culturally specific and contextual social and behaviour change strategies to increase COVID-19-vaccine acceptance in rural settings of the six countries included in this study.

## 2. Materials and Methods

### 2.1. Background of the Study 

This barrier analysis (BA) was conducted in different rural areas of six countries (Bangladesh, India, Myanmar, Kenya, Tanzania, and the DRC) from 7–16 December 2020 to identify behavioural determinants of COVID-19-vaccine acceptance. The study enrolled 452 adults (227 “Acceptors” (Doers) and 225 “Non-acceptors” (Non-doers)) above 18 years of age. Studies from four of the six countries were conducted as part of (and funded through) World Vision’s ENRICH project—a multi-year, multi-country program that aims to improve the health and nutrition status of mothers, new-borns, and children in select rural regions of Bangladesh, Kenya, Myanmar, and Tanzania. The two studies in India and the DRC were conducted as part of World Vision’s privately funded development programs.

In Bangladesh, a March 2021 study found that only 32 percent of respondents were interested in getting a COVID-19 vaccine immediately; 22 percent wanted to get the vaccine after a few weeks, 27 percent after a few months, and 3 percent after one year; while 16 percent did not want the vaccine at all [26]. In Bangladesh, the data collection was conducted in selected communities in Thakurgaon District in the Rangpur Division. 

In India, a January 2021 survey conducted by Local Circles concluded that Indians’ COVID-19-vaccine hesitancy remained unchanged in November and December 2020, with only 41% maintaining that they planned to get the vaccine [27]. In India, data collection was conducted in selected communities in Andhra Pradesh, Madhya Pradesh, and Assam states. By December 2021, this had improved with 81.3% of unvaccinated individuals reporting that they will get a vaccine when available.

In Myanmar, data from the Johns Hopkins Center for Communications Programs estimated that the intent to vaccinate was 80.1% in Myanmar [1] as of December 2021. Rural communities of Thabuang Township of Ayarwaddy Region were selected for the study in Myanmar.

In Kenya, a World Vision study in a rural district found that only 18.9% percent of respondents said they were “extremely likely,” and 21.1% said they were “somewhat likely” to get a COVID-19 vaccine when it was available to them free of cost. Sixty percent of respondents said that they were “somewhat unlikely” (26.5%), were “extremely likely” (17%), or “did not know” (16.5%) if they would get a COVID-19 vaccine (personal communication). In Kenya, data collection was conducted in selected communities in Elgeyo Marakwet County.

In Tanzania, data from the Johns Hopkins Center for Communications Programs estimated that the intent to vaccinate was 60.6% in December 2021 [1]. In Tanzania, data collection was conducted in selected communities in the Shinyanga Region.

In DRC, a survey conducted between August and December 2020 by the Africa Centres for Disease Control and Prevention and the London School of Hygiene and Tropical Medicine found that COVID-19-vaccine acceptance was 59% [28]. In the DRC, data collection was conducted in selected communities in all three health zones of Goma (Karisimbi, Nyiragongo, and Goma). 

### 2.2. Study Approach 

The study used a version of the barrier-analysis rapid formative-research methodology with small data-collection modifications (by phone in Kenya rather than in-person interviews due to COVID-19-related movement restrictions). Unlike most other BA studies, these studies focus on an intended rather than a currently-practiced behaviour (since COVID-19 vaccines were not yet available in most study areas when data were collected). A key feature of the BA is that responses from those doing a behaviour (“Doers” or “Acceptors”) are compared with responses from those who are not (the “Non-doers” or “Non-acceptors”) so that the most-important behavioural determinants can be identified [8]. The other details of the BA approach can be found elsewhere [8,29,30,31,32]. 

### 2.3. Questionnaire 

The questionnaire was divided into two main parts. The first part includes a set of screening questions to identify the participant as either an “Acceptor” or a “Non-acceptor.” In this screening section of the questionnaire, self-reported vaccination-intention questions were included. In order to assess the determinants identified with BA (as referred to Kalam et al. [8]), the second section consisted of a set of open- and closed-ended questions organized by behavioural determinants. The draft questionnaire was based on a template in English that has been used in hundreds of barrier analysis studies throughout the world, including several published studies. The survey teams in five of the six countries translated the questionnaire into the local language and then pretested it. (The team in Tanzania did not translate the questionnaire, but the enumerators, who are fluent in both English and local languages, translated it during their interviews.) The pretesting exercise involved interviews with six Acceptors and six Non-acceptors in each country. The feedback from the pretesting was used to make small modifications to the questionnaire. 

### 2.4. Sample Size 

The BA approach recommends a sample size of 45 Doers (Acceptors) and 45 Non-doers (Non-acceptors) to detect a statistically significant odds ratios of 3.0 or higher, when the alpha error is set to 5% and a power of 80% is desired [8]. The data-collection team approached adult men and women until they reached at least 45 Acceptors and 45 Non-acceptors in each country. In each study site, the enumerators used two-stage cluster sampling to first choose 45 clusters (usually villages) with their likelihood of being chosen to be proportional to their population size and then randomly chose a starting household using the spin-the-bottle technique [33,34]. In the DRC, respondents were chosen in all three health zones of Goma. The 45 Doers and 45 Non-doers were selected from informal worksite places within Goma city using the snowball sampling method, and only one respondent was chosen per worksite. In Kenya, rather than cluster or snowball sampling, telephone lists from beneficiaries enrolled in the ENRICH nutrition project were used to choose respondents randomly. 

### 2.5. Inclusion and Exclusion Criteria 

Acceptors (Doers) were 18 years of age and older and said that they were somewhat likely to, very likely to, or definitely would seek a COVID-19 vaccine if made available to them in the coming month. Non-acceptors (Non-doers) were 18 years of age and older who said that they were somewhat unlikely to, very unlikely to, or definitely would not seek a COVID-19 vaccine, or did not know if they would seek a COVID-19 vaccine if it were made available to them in the coming month. Respondents were excluded if they were under 18 years of age, had never heard of COVID-19, would not say to what degree they were likely or unlikely to seek a COVID-19 vaccine, or did not want to participate in the study.

### 2.6. Data Collection 

Data collectors from the ENRICH project in Bangladesh, Kenya, Myanmar, and Tanzania, and other World Vision staff in India and DRC were trained virtually on the data collection tool and the study methodology. The tool was tested before using it for the actual study. A paper-based questionnaire was used to collect data in all countries but Kenya. Due to the COVID-19 restrictions and travel ban at the time of this study, the Kenya team conducted the interviews over the phone. 

### 2.7. Data Analysis

Data analysis performed to identify the most-important behavioural determinants of COVID-19-vaccine hesitancy in each country used the BA data-tabulation sheet, which generates *p*-values (related to differences between Acceptors and Non-acceptors), estimated relative risks (ERR), and odds ratios (with confidence intervals). In order to rank ERRs when they could not be calculated—or were infinite—due to zero cells, we changed the zero cell to 1 in order to get an estimation of the ERR. Where this was done, rather than providing an exact ERR, such as ERR = 8.2, we expressed it as “ERR > 8.2”. Where ERRs were less than one, we expressed them as their reciprocal (1/ERR) to facilitate ranking and comparison. All findings presented below were at the *p* < 0.05, with many being at the *p* < 0.01, level or lower.

### 2.8. Ethics Statement 

The study was conducted in accordance with the Declaration of Helsinki. The respondents were informed of the study objectives prior to taking part in the interviews, and all respondents were asked to give their written consent before they were interviewed. All participants had the right to withdraw from the study at any moment without jeopardizing their access to any services. The study presented no additional risks to participants. Since this study used a formative research tool regularly used in routine project activities, and since no risks to participants from participating in the study were identified, IRB approval was not sought. Identifying data such as names, dates of birth, and addresses were not collected from the study participants. 

## 3. Results

Based on the ERR value, the study revealed a very high association (ERR or 1/ERR of 8.0 or greater) between responses regarding eight behavioural determinants and vaccine acceptance in Bangladesh, Kenya, Tanzania, and the DRC. These determinants were perceived social norms, perceived positive consequences, perceived negative consequences, perceived risk of getting COVID-19, perceived severity of COVID-19, trust in COVID-19 vaccines, expected access to vaccines, and perceived safety of COVID-19 vaccines. A high level of association (ERR or its reciprocal = 4.0–7.9) on responses regarding eleven of the behavioural determinants was found: perceived self-efficacy, perceived social norms, trust in COVID-19 vaccines, trust in leaders’ COVID-19 information, expected access to COVID-19 vaccines, perceived divine will, perceived action efficacy, perceived positive consequences, perceived negative consequences, perceived severity, and culture (e.g., cultural and religious reasons people plan to not get vaccinated). The findings are presented below, organized under each behavioural determinant (Table 1). Please refer to Appendix A for full results.

### 3.1. Perceived Social Norms

Figure 1 shows the results on perceived social norms. The study showed that close family members, friends, religious leaders, and political and social leaders are key in influencing people’s decision to get a COVID-19 vaccine. A strong majority of Acceptors believed that most of their close family and friends would get a COVID-19 vaccine in Bangladesh (100%), Myanmar (100%), Kenya (91%), Tanzania (62%), and India (62%), whereas Non-acceptors were much less likely to believe that (38%, 84%, 36%, 0%, and 22%, respectively). Acceptors were also more likely to say that most of their community leaders and religious leaders want them to get a COVID-19 vaccine in four of the five countries (all but the DRC) where it was assessed. In four of the five countries where it was assessed, Acceptors were also more likely to say they would get a COVID-19 vaccine if a health worker recommended it. 

Related to social norms, in all four countries where this was assessed, Acceptors were more likely to say that they were very or somewhat likely to get a COVID-19 vaccine if a doctor or nurse recommended it. Conversely, Non-acceptors in these four countries were more likely to say they are not likely to get the vaccine if a doctor or nurse recommends it (This question was not asked in Kenya or India).

### 3.2. Trust in COVID-19 Vaccines 

When asked whether they trust the COVID-19 vaccines, Acceptors were more likely to say that they “trust them a lot” or “trust them a moderate amount” in four of the five countries where it was assessed. Non-acceptors were more likely to say that they “they have no trust at all” or only “trust them a little” in all five countries that assessed this (Figure 2). 

The correlation between lack of trust in the vaccine and being a Non-acceptor was very high, especially in Kenya (ERR = 17.9) and Tanzania (ERR > 12).

### 3.3. Perceived Action Efficacy of COVID-19 Vaccines

To assess the perceived action efficacy (i.e., does a respondent think the vaccine will work as intended to protect him or her from COVID-19 disease), Acceptors were more likely to say that they would not be likely at all to get COVID-19 disease after vaccination in Bangladesh, Kenya, and Tanzania (Figure 3). Conversely, Non-acceptors were more likely to say that they would be somewhat or very likely to get COVID-19 even after they were vaccinated in Bangladesh and the DRC. Despite being a vaccine acceptor, Acceptors in the DRC were more likely to say that they were still “somewhat likely” to get COVID-19 after being vaccinated than Non-acceptors. 

### 3.4. Safety of Getting a COVID-19 Vaccine

Regarding the safety of COVID-19 vaccines, Acceptors in five countries (except India) were more likely to say that they are “very safe” or “mostly safe.” Conversely, Non-acceptors in all five countries were more likely to say that COVID-19 vaccines are “not safe at all” (Figure 4). 

### 3.5. Trust in Sources of Information on the Safety of COVID-19 Vaccines

Acceptors in five of the six countries were more likely to say that they would have a very or somewhat high level of trust in the information that government representatives and politicians provide on the safety and effectiveness of COVID-19 vaccines (Figure 5). In India, paradoxically, Acceptors were more likely to say that they had a very low level of trust in the COVID-19-vaccine info provided by these officials. Non-acceptors in Bangladesh, Myanmar, and the DRC were more likely to have a very or somewhat low level of trust in COVID-19-vaccine information provided by the government or politicians than Non-acceptors. Similarly, Non-acceptors from India, Myanmar, and Tanzania were more likely to say that they have a “very low” or “somewhat low” level of trust in COVID-19 information provided by government representatives or politicians. 

Acceptors in five of the six countries (except India) were more likely to say that they would have a very or somewhat-high level of trust in the information that religious leaders provide on the safety and effectiveness of COVID-19 vaccines (Figure 5). Conversely, Non-acceptors from five of the six countries (all but Kenya) said that they had a somewhat- or very low trust in this information from religious leaders. 

### 3.6. Perceived Positive and Negative Consequences (Advantages and Disadvantages) of Getting a COVID-19 Vaccine

When asked about advantages of vaccination with a COVID-19 vaccine, Acceptors were more likely to cite prevention of COVID-19 as an advantage in five of the six countries (Figure 6). Acceptors in Bangladesh and Myanmar were more likely to say they “won’t transmit COVID-19 to others.” Lifting of travel restrictions was mentioned by Myanmar Acceptors, while the Tanzanian Acceptors were more likely to mention being able to work and earn an income and to reduce the COVID-19 death rate. Non-acceptors were more likely to say that there were no advantages in four of the six countries (all but Kenya and Myanmar). 

When asked about the disadvantages of vaccination with a COVID-19 vaccine, negative or serious side effects were mentioned more often by Non-acceptors in Myanmar (Figure 6). Interestingly, negative or serious side effects were mentioned more by Acceptors in both India and DRC. In the DRC, Non-acceptors were more likely to mention both death and getting other serious diseases as a disadvantage of getting a COVID-19 vaccine. In Tanzania, Non-acceptors were more likely to mention impotence. In Tanzania, Acceptors were more likely to mention a relatively mild side effect, fever. 

### 3.7. Perceived Susceptibility of Getting COVID-19 

When assessing perceived susceptibility to COVID-19, and in line with the HBM, in Bangladesh, India and the DRC, Acceptors were found to be more likely to believe that more people have had COVID-19 (Figure 7a). Interestingly, in Tanzania, Acceptors were more likely to say that no one has had COVID-19 in their community, while Non-acceptors were much more likely to say that “very few people” have had COVID-19. When asked about the likelihood of someone in their household getting COVID-19 over the next three months, Acceptors in four of the six countries (Bangladesh, India, the DRC, and Tanzania) were all more likely to say that the likelihood was very or somewhat likely. 

Conversely, Non-acceptors in Bangladesh, the DRC, and Tanzania were more likely (than Non-acceptors) to say that it was not likely at all that they or someone in their household would get COVID-19 in the next three months (Figure 7a,b). Acceptors were also more likely to say that that they were moderately or very concerned about getting COVID-19 in the next three months in the DRC, Bangladesh, and Myanmar. 

### 3.8. Perceived Severity of COVID-19 

When assessing perceived severity, and in line with the HBM, Acceptors were more likely to believe that it would be very serious if they or someone in their household contracted COVID-19 in four of the six countries (Bangladesh, Kenya, the DRC, and Tanzania), and conversely, Non-acceptors in India were more likely to believe that COVID-19 was not serious at all (Figure 8). 

### 3.9. Perceived Access to COVID-19 Vaccines 

In this study, self-reported distance and waiting times in queue were used as a means of measuring perceived access to vaccine. In Bangladesh, India, Myanmar, and Tanzania, Acceptors were more likely to say that they believe vaccines would be available 30 min from their home (Figure 9). In the DRC, the question was modified to ask how difficult it would be to get to the site where vaccines are normally provided. The DRC study found that Acceptors were more likely to say that it would not be difficult at all, and Non-acceptors were more likely to say that it would be very difficult. Regarding the question on the expected queue time, Acceptors were more likely to say either 0–30 min or 60–90 min (from categories of 0–30 min, 31–60 min, 60–90 min, 1.5–2 h, or 2–3 h) than Non-acceptors in Bangladesh. In Tanzania, Acceptors were more likely to say 1.5 h or more, while Non-acceptors expected queue times of 31–60 min. 

### 3.10. Perceived Self-Efficacy: What Might Make It Easier to Get a COVID-19 Vaccine

Regarding perceived self-efficacy (framed as what might make it easier to get a COVID-19 vaccine), Acceptors were more likely to mention providing the vaccine close to their homes (India); providing it through satellite clinics, primary health-care sub-centres, and health facilities and avoiding stock-outs (Bangladesh, India and Myanmar, and Kenya, respectively); using convincing and clear information on the vaccines and their effects (the DRC); and knowing that COVID-19 is dangerous and offering it free of charge (Tanzania) (Figure 10). 

When asked what might make it difficult for them to get a COVID-19 vaccine, Non-acceptors were more likely than Acceptors to say the lack of information or documentation and being too time-consuming (India); the cost of transportation and having a prolonged illness (Myanmar); the uncertainty about COVID-19 being real (DRC); and not trusting COVID-19 vaccination or not having COVID-19 cases in their community (Tanzania) (Figure 10b). 

### 3.11. Perceived Divine Will 

When assessing perceived divine will, Acceptors from five of the six countries (all except India) were more likely to believe that a deity (God, Allah, or the gods) approves of getting a COVID-19 vaccine (Figure 11). In India, Acceptors were more likely to say that a deity does not approve. 

Respondents were also asked whether or not they agreed with the statement, “Whether I get COVID-19 or not is purely a matter of God’s will or chance—the actions I take will have little bearing on whether or not I get COVID-19.” Findings for this question varied by country. Non-acceptors from Bangladesh believed (“Agree a lot”) it is God’s will, while Non-acceptors from Tanzania “disagree a lot” that it is God’s will or chance that determines whether or not they will get COVID-19. Surprisingly, Acceptors in Bangladesh (80% of Acceptors vs. 27% of Non-acceptors) and in Tanzania (38% of Acceptors vs. 13% of Non-acceptors) believed that it is God’s will or chance whether they get COVID-19 or not. 

## 4. Discussion

An earlier worldwide scoping review by Biswas [35] of studies published between February 2020 and February 2021 found that the most commonly reported determinants affecting COVID-19-vaccination intention included some the same determinants found in this study: perceived vaccine efficacy, vaccine side effects, trust/mistrust in healthcare, religious beliefs, and trust in information sources. This study identified important additional determinants associated with vaccine acceptance in the six study countries: perceived social norms (both descriptive and injunctive norms), perceived self-efficacy, perceived susceptibility to getting COVID-19, perceived severity of the disease itself, and other perceived positive and negative consequences of getting a COVID-19 vaccine. We agree with Biswas and co-authors that “The underlying factors of vaccine hesitancy are complex and context-specific, varying across time and socio-demographic variables”; so, periodic assessment of determinants in every country where vaccine hesitancy is an issue is merited, and reporting of those country-specific factors in the scientific literature is useful. In this six-country study, the determinants mentioned below were found to have the largest associations with COVID-19-vaccination acceptance. 

***Perceived social norms:*** The study revealed the powerful effect of perceived social norms on vaccination behaviour. Community leaders, religious leaders, close family members, and friends were reported to be very influential in mobilizing communities for COVID-19 vaccination. This lends support to other studies that have found that perceived social norms are positively associated with people’s intentions to get vaccinated, including vaccination with a COVID-19 vaccine, and the importance they give to getting vaccines [8,36,37]. As vaccines are made available to more and more people in these countries, it will be important that people see or hear that most people around them are planning to get a COVID-19 vaccine. The belief that “most people I know” and “most of my close friends and family” are going to get vaccinated for COVID-19 was one of the strongest predictors of being a vaccine Acceptor. A strong majority—and, in Bangladesh and Myanmar, the totality—of Acceptors in all six countries believed that most of their close family and friends would get a vaccine. 

Perceptions of the degree to which community and religious leaders support COVID-19 vaccination were also significant and demonstrate the importance of mobilizing community and religious leaders to be involved in efforts to increase COVID-19 vaccine acceptance, which is aligned with other studies [38,39]. Acceptors were much more likely to say that these leaders approve of their getting a vaccine in Bangladesh (100%), Myanmar (100%), Tanzania (100%), and Kenya (96%), whereas a much smaller proportion of Non-acceptors reported this (38%, 80%, 29%, and 53% respectively). Acceptors were found to be much more likely to have high levels of trust in government representatives, political leaders, and religious leaders than Non-acceptors. 

***Trust in vaccines and vaccine messaging:*** Accurate and reliable information on the safety and effectiveness of COVID-19 vaccines will need to be made available to populations in these countries through a variety of trusted channels and leaders to increase COVID-19-vaccine acceptance [40,41,42]. A lack of trust in COVID-19 vaccines was common among Non-acceptors in all five countries where it was assessed (Bangladesh, Kenya, Myanmar, the DRC, and Tanzania). Acceptors in all countries where perceptions of safety were assessed were more likely to believe that it would be very or mostly safe for them to get a vaccine. Acceptors in Bangladesh, Kenya, the DRC, and Tanzania were also much more likely to believe that COVID-19 vaccines work (i.e., they will not get COVID-19 after being vaccinated). Information will need to be disseminated to counter (but not repeat) common myths about the vaccines’ safety and effectiveness, such as beliefs in the DRC and Myanmar that the vaccines have negative and potentially deadly side effects and beliefs about the vaccines causing impotence in Tanzania. Perceived safety and trust in COVID-19 vaccines and in those promoting the vaccines (e.g., health workers and government workers) have been found to be important determinants of COVID-19 vaccination in other countries [42].

***Perceived positive consequences of COVID-19 vaccines:*** As expected, prevention of COVID-19 was mentioned as an advantage by Acceptors (more often than Non-acceptors) in five of the six study countries (all but Kenya, where that advantage was mentioned by high proportions of both Acceptors and Non-acceptors). However, the findings on other benefits of COVID-19 vaccination—such as being able to travel in Myanmar and being able to work and earn an income in Tanzania—should be leveraged to increase vaccine acceptance in those countries. 

***Perceived severity of—and susceptibility to—COVID-19:*** As predicted by the HBM and other current COVID-19-vaccine-hesitancy studies [43], beliefs about COVID-19 and its prevalence and severity were strongly associated with vaccine acceptance and should be taken into account when developing messages and activities to promote COVID-19 vaccines. In Bangladesh, India, and the DRC, Acceptors were more likely to believe that more people had been infected with COVID-19. In Bangladesh, India, the DRC, and Tanzania, Acceptors were more likely to believe that it was likely that someone in their household would get COVID-19 in the next three months. In Myanmar and the DRC, concern about getting COVID-19 was much higher among Acceptors than Non-acceptors. Acceptors in Bangladesh, Kenya, the DRC, and Tanzania were much more likely to believe that COVID-19 was very serious, and in India, Non-acceptors were more likely to say that COVID-19 was not serious at all. 

***Access to COVID-19 vaccines:*** Even before vaccines began rolling out in many communities, people’s beliefs about access—such as expected queue times and the difficulty of getting to places where vaccines are normally available—were found to be highly associated with vaccine acceptance [39]. For example, Acceptors in Bangladesh, India, Myanmar, and Tanzania were much more likely to say that they expect that COVID-19 vaccines will be available within 30 min of their home, and Acceptors in the DRC (who were asked a similar but not identical question) were more likely to say that it would not be difficult at all to get to a vaccination site. Regarding expected queue times, countries varied. In Bangladesh, Acceptors were more likely to expect shorter queue times, while in Tanzania, Acceptors were more likely to expect longer queue times (of 1.5 h or more) than Non-acceptors. 

***Perceived self-efficacy:*** The findings regarding what makes COVID-19 vaccination easier or more difficult can provide clues to communication and vaccination strategies. Acceptors were more likely to mention access-related issues that could make vaccines easier to get, such as providing them by NGOs, through satellite clinics and sub-centres, and assuring vaccines are available on time, in the right places, without stock-outs, and free of charge. Providing clear and convincing information was also mentioned. Similarly, Non-acceptors were more likely to mention costs in money and time, having prolonged illnesses, lack of information, and trust in the vaccines as things that would make getting a COVID-19 vaccine more difficult. 

***Perceived divine will and faith beliefs:*** There is a growing body of evidence on the impact of faith leaders and faith beliefs on health and other behaviours [8,39,44]. In April 2020, the WHO concluded that “religious leaders, faith-based organizations, and faith communities can play a major role in saving lives and reducing illness related to COVID-19” [44]. This study found that Acceptors were much more likely to believe that a deity (God, Allah, or the gods) approved of them getting a COVID-19 vaccine in all six countries. However, the correlation between personal agency and vaccine acceptance was mixed. In Tanzania, Acceptors were more likely to agree that getting COVID-19 was “purely a matter of God’s will or chance,” while Non-acceptors were much more likely to strongly disagree with that statement; in Bangladesh, Non-acceptors were much more likely to say that they strongly agreed with that statement (than Acceptors).

***Limitations:*** There are several limitations of this study. First, this study was conducted during an earlier phase of the COVID-19 pandemic, which had a different epidemiological context from the current phase. Determinants and drivers of vaccine acceptance may well have shifted since that earlier phase, and updated studies are needed to confirm how the drivers of COVID-19-vaccination acceptance have changed over time. Second, this study was conducted in less than 45 communities within each country, so it is not an exhaustive national sampling of rural areas.

## 5. Conclusions

In conclusion, COVID-19-vaccine hesitancy is a roadblock that many countries are facing along the road to herd immunity and an end to the pandemic. This study shows that there are many behavioural determinants associated with vaccine acceptance that need to be regularly explored through formative research to understand better which messages and activities should be used to counter vaccine hesitancy and refusal. These determinants can vary from country to country, and from phase to phase of a pandemic, but certain ones (e.g., perceived social norms, perceived severity of COVID-19, and perceived divine will) may be reliably found to be important in many countries. 

National and local plans for COVID-19 vaccination should include the participation of community and faith leaders, health workers (including community health workers), and others to mobilize communities for COVID-19 prevention, case detection, and referral, including promotion of COVID-19 vaccination. It is crucial to empower these leaders with trustworthy information on COVID-19 and the safety and effectiveness of COVID-19 vaccines, but also on the probable and varied behavioural determinants of COVID-19 vaccination in their area. The decision to seek or not seek vaccination is not solely reliant on one’s views concerning the safety and effectiveness of vaccines or other beliefs about vaccines (e.g., side effects and myths). Tools and resources for these leaders and workers should be contextualized using data on the determinants of vaccine hesitancy in a given area or country. Strategies for COVID-19 prevention (including but not limited to COVID-19 vaccination) may need to address many of these determinants to be effective.

## Figures and Tables

**Figure 1 vaccines-10-00214-f001:**
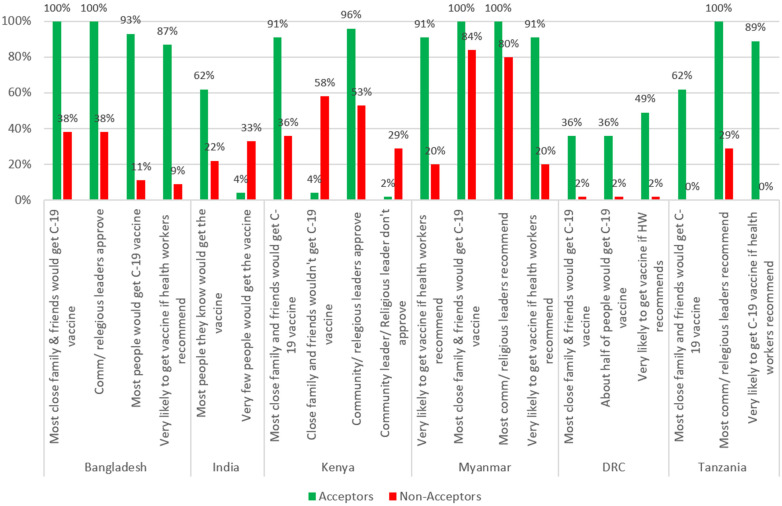
Perceived social norms on COVID-19-vaccine hesitancy.

**Figure 2 vaccines-10-00214-f002:**
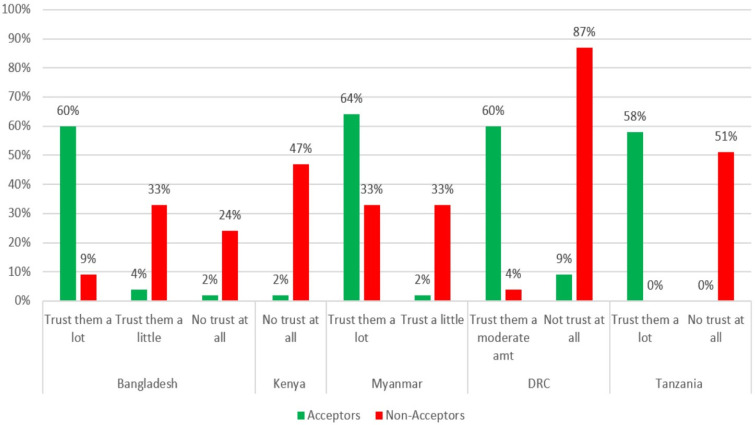
Acceptors’ and Non-acceptors’ trust in COVID-19 vaccine by country.

**Figure 3 vaccines-10-00214-f003:**
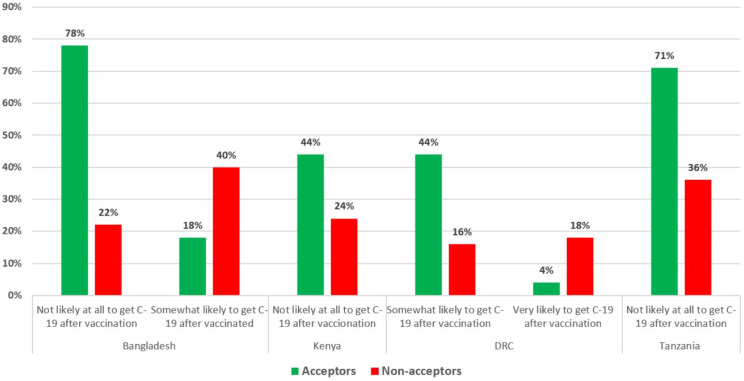
Acceptors vs. Non-acceptors: Perceived action efficacy of COVID-19 vaccines by country.

**Figure 4 vaccines-10-00214-f004:**
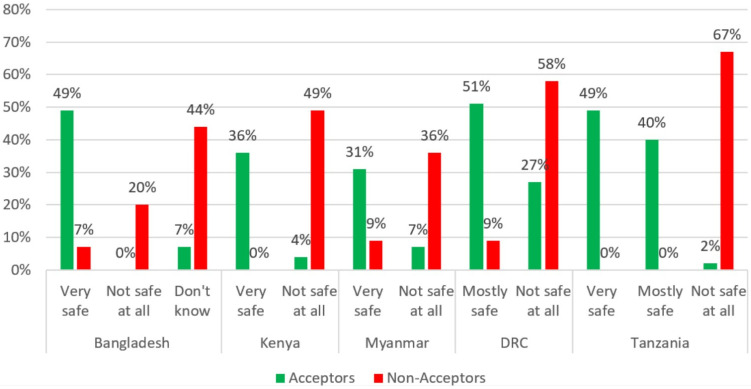
Acceptors’ and Non-acceptors’ beliefs on the safety of getting a COVID-19 vaccine by country.

**Figure 5 vaccines-10-00214-f005:**
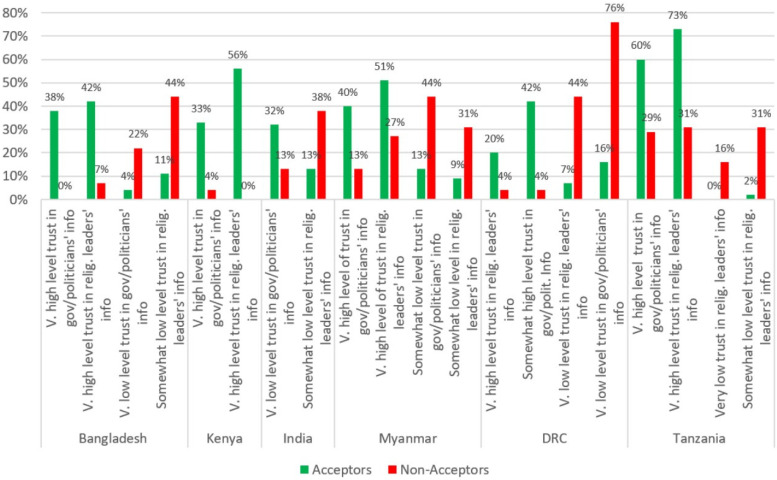
Acceptors’ and Non-acceptors’ trust level in COVID-19 safety and effectiveness information by source of information.

**Figure 6 vaccines-10-00214-f006:**
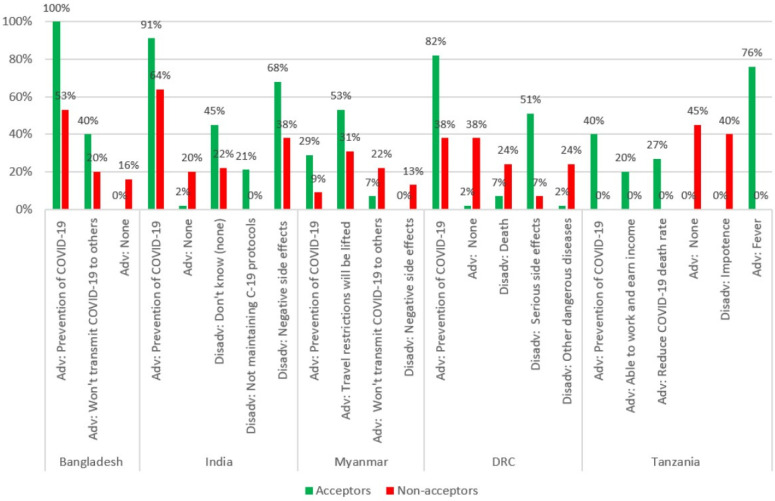
Perceived positive and negative consequences (advantages and disadvantages) of getting a COVID-19 vaccine by country.

**Figure 7 vaccines-10-00214-f007:**
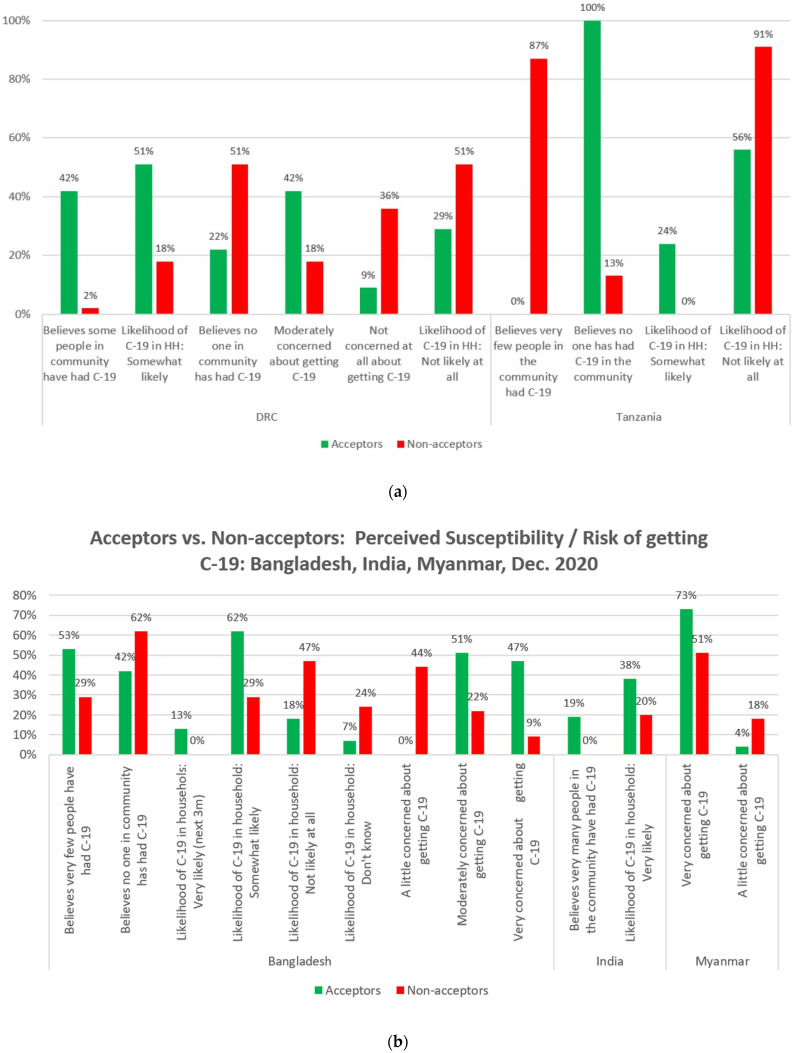
(**a**): Perceived susceptibility/risk of getting COVID-19 in DRC and Tanzania. (**b**): Perceived susceptibility/risk of getting COVID-19 in Bangladesh, India, and Myanmar.

**Figure 8 vaccines-10-00214-f008:**
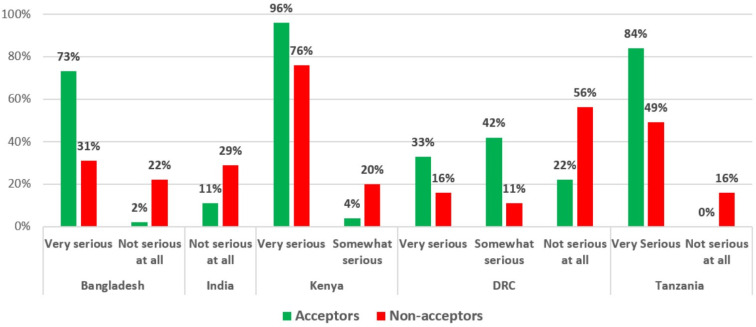
Perceived severity of COVID-19 by country.

**Figure 9 vaccines-10-00214-f009:**
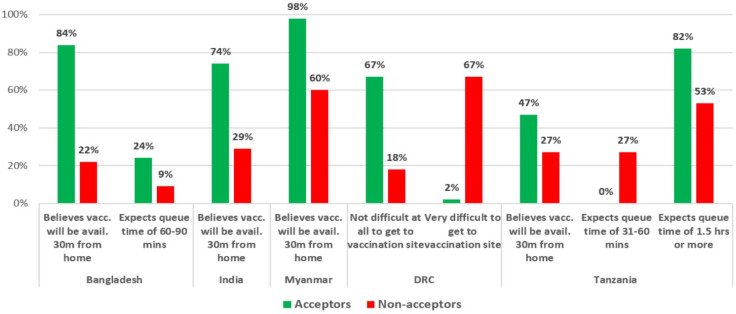
Perceived access to COVID-19 vaccines by country.

**Figure 10 vaccines-10-00214-f010:**
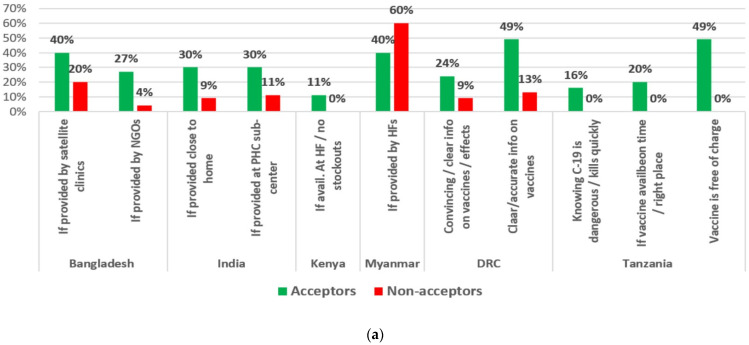
(**a**): What might make COVID-19 vaccination easier? (**b**): What might make COVID-19 vaccination difficult?

**Figure 11 vaccines-10-00214-f011:**
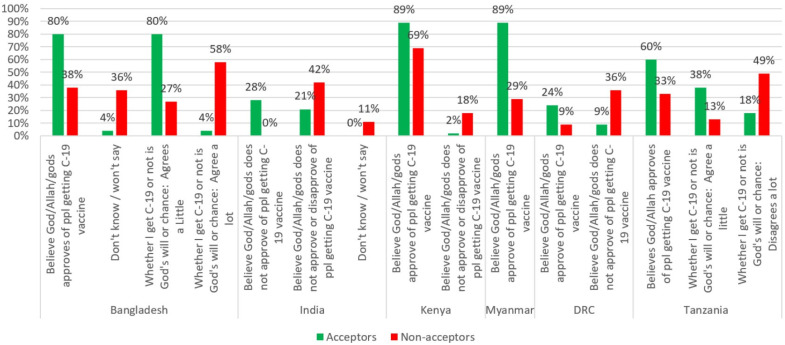
Perceived divine will regarding COVID-19 vaccination by country.

**Table 1 vaccines-10-00214-t001:** Reference: degree of association between behavioural determinants and COVID-19-vaccine acceptance by country.

Determinants	Bangladesh	India	Kenya	Myanmar	DRC	Tanzania
Vaccine acceptance (est. at time of study)	64%	41%	40%	75%	59%	61%
Perceived self-efficacy	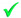	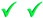	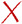	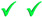	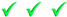	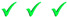
Perceived social norms	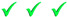	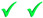	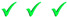	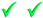	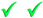	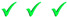
Perceived access	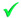	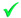	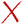	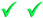	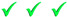	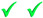
Perceived positive consequences	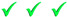	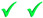	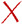	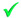	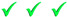	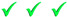
Perceived negative consequences	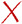	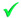	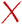	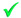	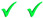	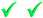
Perc. susceptibility/risk (of getting COVID-19)	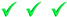	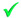	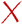	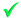	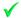	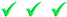
Perceived severity (of COVID-19)	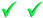	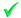	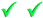	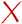	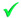	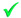
Perceived action efficacy	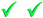	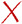	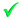	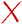	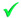	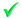
Perceived divine will	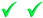	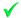	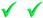	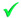	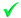	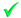
Trust (in information)	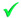	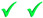	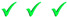	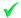	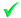	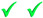
Trust (in vaccines)	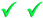	Not asked	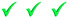	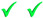	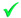	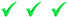
Safety	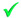	Not asked	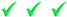	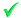	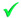	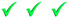
Cultural (taboos, etc.)	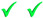	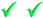	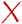	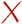	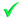	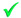
Other correlate (e.g., educational level)	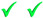	Not asked	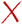	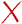	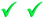	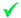

One check: Estimated Relative Risk (ERR) < 4; two checks: ERR 4.0–7.9; three checks: ERR > 8.0 or greater; ERR shows the level of association between the determinant and vaccine acceptance.

## Data Availability

All data related to this study are reported in the main manuscript and its Appendix A.

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
