# Peer review of "Behavioural Determinants of COVID-19-Vaccine Acceptance in Rural Areas of Six Lower- and Middle-Income Countries"

_vaccines, 2022, doi:10.3390/vaccines10020214_

Round 1

Reviewer 1 Report

The manuscript is very well-written, describing the findings of a study on behavioral determinants of COVID-19 vaccine acceptance in six countries of Asia and Africa. The methodology, results, and their subsequent discussion are clearly presented, and the work is potentially useful to guide health agents to promote better communication to inform and educate the public about the importance of vaccination. In a future study, my suggestion is to include questions about the perceived efficacy (vaccines are not 100% efficient) and the mechanism of the vaccination process (vaccines are not meant to be individual shields, but instead to act collectively to reduce  virus circulation and consequently avoid potential mutations), as those are crucial questions that are not adequately dealt by the media.  

Minor comment:

313 - Define "HH" (appearing in Figure 7a).

Reviewer 2 Report

Thank you for asking me to review this article. COVID-19 is an ongoing pandemic that has resulted in global health, economic and social crises. Understanding the determinants of vaccination acceptance and coverage is a useful tool for implementing strategic measures aimed at improving patient compliance with vaccination with particular reference to anti-COVID-19 vaccination.

In this context, the paper under evaluation is aimed at identifying behavioural determinants of COVID-19 vaccine acceptance and provide recommendations to improve the immunization in six lower-and-middle income countries.

The subject under study is certainly very important, especially in the historical period we are experiencing. The article presents interesting results but, given the organization of the contents and the description of the same, the manuscript cannot be published in its current form.

Title: it is overstated since it is carried on in six schools enrolling a sample of people.

Introduction: The authors should better clear what is the gap in the literature that is filled with this study. The paper is too much targeted at a local level. Therefore, since the vaccination campaign success is influenced by people’s acceptance of these vaccines, this issue must be discussed in the introduction reporting level of acceptance also in other students population (refer to article with doi: 10.3390/vaccines9060638). Finally, the Authors must explicit what is the potential contribution of the study to the literature and its implications.

Methods: The enrolment procedure must be better specified, is seems a little confusing. Who was involved in the survey and how? How did the authors choose the way used to enroll their sample? How did they avoid the selection bias? What is the reference population? All population from the six countries? what is the minimum sample related to the reference population and the power of the study? The Authors report that “The BA approach recommends a sample size of 45 Doers (Acceptors) and 45 Non-doers (Non-acceptors) to detect statistically significant Odds Ratios”, this makes no sence, the significance cannot be a priori assumed and dipends from different parameters.

No mention to a validation process of the questionnaire is reported. What about face validity, reliability and intelligibility?

Statistical analysis: I suggest to insert a measure of the magnitude of the effect for the comparisons, include an effect sizes analysis.

Ethical Issue: although an anonymous questionnaire is used, an ethical approval is necessary. An ethical committee should approve the study protocol, and a reference number should be reported.

Discussion: It is sometimes redundant, it should be reorganized emphasizing the contribution of the study to the literature, and overall the limits (the study was performed in a differente epidemiological context!). The authors report the results but it is not clear their practical impact. The already demonstrated impact of information campaign on COVID-19 vaccine acceptance should be clearly discussed (refer to article with doi: 10.3390/vaccines9060638).

Round 2

Reviewer 2 Report

The article was improved according to comments provided, the article is now suitable for publication